# Gender Differences of University Students in the Online Teaching Quality and Psychological Profile during the COVID-19 Pandemic

**DOI:** 10.3390/ijerph192214729

**Published:** 2022-11-09

**Authors:** Simone Nomie-Sato, Emilia Condes Moreno, Adriana Rico Villanueva, Pascual Chiarella, Jose Francisco Tornero-Aguilera, Ana Isabel Beltrán-Velasco, Vicente Javier Clemente-Suárez

**Affiliations:** 1Faculty of Sports Sciences, Universidad Europea de Madrid, 28670 Madrid, Spain; 2Faculty of Biomedical and Health Sciences, Universidad Europea de Madrid, 28670 Madrid, Spain; 3Centro de Opinión Pública, Universidad del Valle de México, Mexico City 14370, Mexico; 4School of Health Sciences, Universidad Peruana de Ciencias Aplicadas, Lima 15023, Peru; 5Studies Centre in Applied Combat (CESCA), 45007 Toledo, Spain; 6Department of Psychology, Faculty of Life and Natural Sciences, University of Nebrija, 28240 Madrid, Spain

**Keywords:** COVID-19, online teaching, hybrid education, mental health, gender differences

## Abstract

With the arrival of COVID-19, educational systems have had to adapt to the social and health situation immediately. This led to the appearance of the asynchronous teaching model. Throughout the pandemic, at the educational level, we can distinguish three phases, eminently online, hybrid, and finally, face-to-face. However, the perception of educational quality in these three educational moments, taking into account the psychometric profile and gender, has not been studied. Thus, 1093 university students from Ibero-American countries were analyzed. Through a questionnaire, demographic, academic, and psychological variables were analyzed at three moments during the evolution of the pandemic. Data suggest that, during the lockdown phase, while teaching was eminently online, students presented higher levels of stress and higher difficulty of learning; class attendance, convenience, preferred method of learning, grading score, and motivation were lower, compared to other phases of teaching (hybrid and face-to-face). During this period, females presented higher stress levels than males, as well as higher levels of anxiety and loneliness, without gender differences among the other studied variables. During the hybrid and face-to-face phases, male students presented higher values in the results of difficulty learning and demanding activities. No differences were seen regarding motivation, synchronous class attendance, learning level, grades, convenience, or preferred learning method. The results from the present study suggest that, despite the effect of the pandemic on mental health, asynchronous education is postulated as an effective teaching–learning alternative. Yet, a special focus should be given to female students.

## 1. Introduction

SARS-CoV-2 rapidly spread around the world, leading to the COVID-19 pandemic, forcing governments to adopt strict measures, and sending billions of people into lockdown [1]. A state of emergency was declared by the World Health Organization (WHO), and in synchrony with the countries of the world, measures to contain the virus and maintain medical systems without collapse were adopted [2]. Measures such as the immediate closure of public spaces, such as restaurants, cafes, and gyms, as well as in sports and educational centers, were taken [3]. After the slowing of the contagion curve, the gradual opening of these spaces made the use of face masks mandatory [4]. In this process, one of the most affected sectors was education, from primary to higher education, adopting intensive measures to prevent and protect all students and staff members from highly infectious diseases [5,6]. Regardless of these measures, the impact, according to UNESCO data, was a total of 1.3 billion learners who were not able to attend school or university [7].

In the case of university institutions, during the academic years 20/21 and 21/22, universities have proposed alternatives to the face-to-face model to an emerging online modality and have forced an emergency readaptation in all educational stages. Universities started in a synchronous format, that is, with professors and students meeting at the same time in classes through an online platform, and the practical activities were postponed or substituted when possible. In the first six months, most of the universities from Ibero-American countries switched to this 100% online modality [8].

The implementation of this new educational system and model, however, has had an impact on students, teachers, and institutions. Students present difficulties in the learning process, increase the workload for teachers, and highlight the importance of technological resources and virtual infrastructures of institutions [9]. Indeed, most university students and professors prefer face-to-face classes in a physical environment, since it is perceived as a better interaction or brainstorming discussion during the class than in online teaching [10]. Previous authors suggested adaptation difficulties and resistance to a technological adaptation of professors to online teaching, due to inexperience, new resources, time, or feeling less interaction and discussion in online classroom teaching [8,9,10,11]. Furthermore, professors reported increased time to explain and greater issues in the teaching and learning process [12,13]. In this line, given the rapid process of the virus and the change to online teaching, there was no time for training professors in the pedagogical methodologies designed online [14]. Thus, it is important to consider efficient training programs for the success of the teaching–learning process and mitigation or reduction of teacher stress and burnout [15].

Furthermore, the delivery of online education has highlighted the difficulties of some countries with lower economic potential, which struggled with the availability of internet access at all homes [16], and low-income families who cannot afford to purchase a proper device for their children [17]. However, students with social, cultural, geographical, and economic constraints, including those with low proficiencies in English and technological skills, are experiencing disconnect and disengagement [18]. There is a “digital division”; thus, policies and strategies need to be formulated, exploring solutions to their respective context and enabling the continuation of online teaching and learning activities. Infrastructural and digital development amongst pedagogic and economic support is essential in collaboration with governments, universities, and communities [12,13].

All this translates into a strong impact on mental health, creating a large level of stress among the university community, professors, and students. This stress may lead to unfavorable effects on the learning and psychological health of students. In a systematic review, we can identify many articles that studied the impact of the pandemic on the mental health of students and professors, showing a high prevalence of anxiety, depression, and stress, with a large variation between studies [19,20,21,22,23]. On the other hand, it is well-known that gender is an element that can be a determinant of people’s health. In this line, women tend to present more frequent pathologies related to alterations in mood, such as anxiety disorders and depression. However, in men, it is more common to find pathologies related to interactions with the environment, such as substance use and the expression of disorders related to difficulties in relationships with others [24,25,26,27,28]. Another determining factor is the social environment, since, in environments where women are educated with a marked difference between genders, more serious consequences appear. In this sense, the months in which the pandemic has damaged affective and attachment relationships with family members, together with a clear decrease in social relationships, puts women at clear risk of suffering a mental disease [29,30,31,32,33,34]. Research in this field has been able to confirm that significant correlations can be established between gender differences and social determinants, concerning the appearance of mental illnesses. Gender differences and their consequences on mental well-being can be observed in all countries of the world.

Thus, given the impact in terms of not only educational quality, but also in the mental health of the population, this article shed light on the psychological profile and perception of quality in the teaching–learning processes at the university stage during the three processes of educational transition during COVID-19: online, hybrid, and face-to-face. Likewise, the gender differences that other authors had previously highlighted were considered. The initial hypothesis was that gender differences would be found in online teaching on the perception of quality and psychological profile during the COVID-19 pandemic.

## 2. Materials and Methods

In the current study, 1093 university students from Ibero-American countries were analyzed, aged between 18 and 31 years. Subjects were interviewed via online questionnaire for a period of 6 months, from December 2021 to June 2022. Our inclusion criteria were: enrollment in the current academic year, currently living in Ibero-American countries, and either graduate or postgraduate students from any field/area of expertise. In order to prevent double responses from the same person, students had to include their Student ID, which was required to match with the university database. Furthermore, data were considered strictly confidential. This research complied with the Helsinki declarations on human research and was approved by the University Ethics Committee (CIPI/213006.55). All the participants digitally signed a consented participation, where the aims and procedures of the study were explained. To reach the aim of the present research, a cross-sectional study was developed. The following parameters were analyzed.

### 2.1. Demographic and Biological Information

Participants provided information about gender, age (years), height (cm), weight (kg), body mass index (BMI, Kg/m2), country, and city. In addition, we asked about the environment, digital resources available to attend remote classes during the lockdown period, and the number of cohabitants.

### 2.2. Academic Information

The participants provided information about the academic year, knowledge area, program, level academic (undergraduate or graduate), learning delivery modality (face-to-face, hybrid, or online), average grade before the pandemic period, experience in online teaching environments and digital resources, type of classes (synchronous or asynchronous) during the pandemic period, and availability of recorded classes.

### 2.3. Classes during the Pandemic Period

We analyzed three different moments during the pandemic period, regarding the learning experience under student perception: (a) Lockdown phase/online teaching: all classes were transferred to emergency remote teaching, thus all classes and learning process was delivered online. (b) Hybrid phase: mixed classes between online teaching and face-to-face classes with reduced capacity due to COVID-19 restrictions; and (c) Presential phase/face-to-face: return to classes in person without capacity limit, but with COVID-19 restrictions. In each phase, students had to answer about stress level, motivation, learning level, convenience to learn, grades, work demand, learning difficulties, attendance to synchronous classes, and preference about the received class format, using a Likert scale, where 1 means the lowest and 5 the highest score.

### 2.4. Psychological Factors

We analyzed the students’ perception of how COVID-19 and the effects of the pandemic affected them personally, regarding emotional aspects, the impact on academic activity, and the perception of teaching–learning quality during the health crisis. Data were collected using (a) UCLA loneliness scale [24], composed of 3 items to assess how often a person feels disconnected from others using a Likert scale, where 1 means rarely and 3 frequently; (b) STAI scale: the State–Trait Anxiety Inventory scale [25], composed of 6 items, was applied towards differentiating between the temporary condition of “state anxiety” and the more general and long-standing quality of “trait anxiety” using a Likert scale, where 1 means not at all and 4 very much; and PSS-4: perceived stress scale [26], composed of 4 items to measure the degree to which situations in one’s life are appraised as stressful using a Likert scale, where 0 means never and 4 very often.

### 2.5. Statistical Analysis

For the use of the methodology at the statistical level, as well as the tests used, we have followed the same tests as in previous research [32,34]. Thus, the SPSS statistical package (version 21.0; SPSS, Inc. Chicago, IL, USA) was used to analyze the data. Normality assumptions were checked with a Kolmogorov–Smirnov test. To analyze differences between genders, a *T*-test for independent samples was administered. The level of significance for all the comparisons was set at *p* ≤ 0.05.

## 3. Results

A total of 1093 students completed the online survey. Participants’ characteristics were: mean age of 23.2 ± 6.2 years, BMI (23.8 ± 4.9), 40% males, and 60% females. The participants were resident in Latin American countries (76%) and Europe (24%). The educational levels that they were enrolled in were undergraduate (90%) and postgraduate (10%), whose branches of study were health sciences (45%), social sciences (39%), and higher studies in architecture and engineering (16%). A total of 70% of the participants reported having previous experience with online teaching, 89% declared themselves proficient in the use of digital resources, while only 61% claimed to have their own digital resources for online teaching (WIFI or computer).

As shown in Table 1 and Table 2, gender differences were found in general stress levels during the lockdown, and females presented significantly higher values than male students, as well as higher stress levels in the online classes during the lockdown, STAI—State–Trait Anxiety Inventory scale, UCLA—loneliness scale, and PSS—perceived stress scale.

Furthermore, the male students presented higher values in the results of difficulty learning and demanding activities during the hybrid phase and difficulty learning during the face-to-face phase. This means that the students during this period found greater difficulties in learning, as well as greater difficulty in carrying out the tasks of the degree or postgraduate course.

We did not find statistical differences in motivation, synchronous class attendance, learning level, grades, convenience, or preferred learning method independent of the phase during the COVID-19 period.

## 4. Discussion

This research aimed to analyze the gender differences of university students in online teaching on the perception of quality and psychological profile during the COVID-19 pandemic. The initial hypothesis was completed, since gender differences were found in the perceptions of quality and psychological profiles during the COVID-19 pandemic.

In the present study, differences between males and females were seen on the psychometric profile. Thus, the perception of the situation, the measures taken, the capacity and adaptability to the new scenario, and the different phases may differ. Our data suggest that females had higher stress levels during the confinement phase of the COVID-19 pandemic than males, as well as higher levels of anxiety and loneliness. In this line, authors reported similar results, showing that females have a higher perception of danger from COVID-19 and higher values of anxiety, conscientiousness, neuroticism, and openness to experience than males [27]. This may be explained due to the greater emotional vulnerability of females, as reported by previous authors [28], and the lower stress coping abilities than males [29], consequent with previous studies conducted during COVID-19 in females [30], thus supporting the present results. Furthermore, recent reviews suggest that the female gender is a risk factor for mental health problems, especially during COVID-19 [31,32,33]. Yet, there are no gender differences among the other studied variables during the lockdown; however, it was observed that the average values, compared to other phases (hybrid and face-to-face), were higher for stress level, synchronous class attendance, convenience to learn, difficulty to learn, and grading score, while lower for motivation, perceived learned, and preferred learning method. These high levels of stress and lower motivation are related to psychological factors and social concerns as the outcome of student experiences, feelings of loneliness, fear of a pandemic, worries about health and the health of loved ones, and lack of communication with classmates and relatives [34].

With the improvement of the pandemic scenario, universities were able to return to face-to-face activities with limited capacity, and hybrid teaching (synchronous teaching) was widely used in several countries. Regarding the hybrid phase, male students presented higher values in the results of difficult learning and demanding activities during this phase. Indeed, the authors suggest that the mean scores for females were generally higher than males for all contributing factors toward students’ readiness for hybrid learning; however, the differences between the two tested groups were not significant [35]. It is equally important to consider that there has not been a correct adaptation of contents and methodologies to facilitate teaching. This has caused great difficulties both for students and for the teachers and agents involved in the educational process [36]. The methodologies used during this period have failed to be innovative and to involve students in their learning. Classes have returned to the structure that has been widely criticized, theory classes without practical application that allows students to participate and interact. This requires students at all educational stages to use cognitive resources that are not available in all cases; therefore, students’ performances are damaged [37]. If we talk about hybrid models, they do not seem to have favored the teaching–learning process, since, on many occasions, the individual characteristics of each student must decide between attending in person or online in a reduced period, which is sometimes impossible [38].

After the vaccine and the attenuation of COVID-19 cases, the governments of many countries authorized the opening of universities and the gradual return to face-to-face activities. When it was possible to start face-to-face classes again, it was possible to confirm that the students very positively value being able to go to the educational center. Not only because it is more effective for the learning process, but also because a network of relationships can be generated that facilitates integration and confidence in oneself and colleagues [39,40]. Studies in this line of work showed that there is a significant decrease in the quality of emotional and affective implications, due to the lack of face-to-face communication between students and, therefore, the loss of quality of relationships between people (student–student or student–instructor) in a more solitary online environment that requires a higher level of development [41]. The challenges students faced during this period, such as study demands, time pressure, emotional exhaustion, perceived social support, or student engagement, may explain the influencing mechanism of students’ mental health and this impact on the perceived difficulty to learn during the face-to-face classes. Interestingly, males presented higher values in difficulty learning during the face-to-face phase, which generates certain dissonance by observing the female psychometric profile. Yet, gender-related risk factors for and experiences of burnout and poor mental health remain under-researched and under-reported in the post-pandemic crisis. The educational model change and the return to the traditional model could be an extra stressor for students, being a source of stress that could affect other life aspects, such as personal and interpersonal relations and daily behaviors [27,42,43,44], and having, in this aspect, information access, a direct effect to modulate the stress perception and a fact that would indirectly affect the educational process [45].

### 4.1. Practical Application

After the COVID-19 pandemic, the university authorities should continue to invest in online education to enhance the learning experience. Proper training of professors, regarding digital skills and improved student–teacher interaction must be conducted. For disadvantaged students, the availability of digital infrastructure with proper internet availability and access to technology is essential.

Students are likely to suffer from stress, anxiety, depression, and loneliness, so it is necessary to provide emotional support to students. It is possible to conduct both active and passive mental health interventions, offering, in each case, a style of therapy, either face-to-face or with the use of technological means, such as telephone lines or videoconferences. In this line, women tend to suffer a greater impact on mental health than men; therefore, special attention should be given to this group.

Research in this line could include the recording and analysis of different factors that allow the identification of protective elements of the mental well-being of students at all educational stages. In addition, a preventive action plan could be developed to ensure the correct intervention of mental health support and assistance.

### 4.2. Limitation of the Study and Future Research Lines

Several limitations of the present study should be noted. It should be considered that the recording of the data used in this study was subject to social desirability bias, which is common among interviewees. It could also have been affected by recall bias, since self-reporting requires cognitive tools from the area of memory.

In addition, this research project was conducted at different phases during the pandemic, and data were also collected in different countries and even continents. This means that the progression of the disease was not the same everywhere; therefore, the results in each country may indicate different stages.

However, the results obtained are still relevant, since they allow us to analyze the consequences of the pandemic, concerning the quality of online teaching. Not many studies have been carried out in this sense; therefore, this study provides answers to important questions and allows us to point to future questions, such as determining what are the factors that may be useful for making comparisons between different educational agents or how it has impacted cities of the same country, between countries, and more globally.

## 5. Conclusions

We can conclude that, during the lockdown phase and online teaching, students presented higher levels of stress, class attendance, convenience to learn, difficulty to learn and lower grading scores, motivation, perceived learned, and preferred learning method, compared to other phases of teaching. During this period females presented higher stress levels than males, as well as higher levels of anxiety and loneliness, without gender differences among the other studied variables. During the hybrid and face-to-face phases, male students presented higher values in the results of difficulty learning and demanding activities. No differences were seen regarding motivation, synchronous class attendance, learning level, grades, convenience, or preferred learning method.

Thus, despite the effect of the pandemic on mental health, asynchronous education is postulated as an effective teaching–learning alternative.

## Figures and Tables

**Table 1 ijerph-19-14729-t001:** Differences between genders in the variables of perception of academic quality during the lockdown.

Variable	Female	Male	t	*p*	95% Confidence Interval	
					Lower	Upper
General stress level during lockdown (1–10)	7.6 ± 2.4	6.6 ± 2.7	6.403	0.000	0.697	1.313
Motivation during lockdown (1–5)	2.7 ± 1.2	2.7 ± 1.2	−0.445	0.656	−0.181	0.114
Synchronous class attendance during lockdown (1–5)	3.5 ± 1.5	3.6 ± 1.4	−0.660	0.509	−0.240	0.119
Stress level during lockdown (1–5)	3.4 ± 1.4	3.0 ± 1.3	5.572	0.000	0.307	0.640
Motivation during hybrid phase (1–5)	2.9 ± 1.3	3.1 ± 1.3	−1.799	0.072	−0.309	0.013
Synchronous class attendance during hybrid phase (1–5)	3.1 ± 1.5	3.3 ± 1.4	−1.849	0.065	−0.363	0.011
Stress level during hybrid phase (1–5)	3.0 ± 1.5	2.9 ± 1.3	0.669	0.504	−0.114	0.231
Motivation during face-to-face phase (1–5)	3.1 ± 1.6	3.1 ± 1.6	−0.518	0.605	−0.243	0.141
Attendance to face-to-face classes (1–5)	3.0 ± 1.7	3.2 ± 1.6	−1.176	0.240	−0.329	0.082
Stress level during of face-to-face phase (1–5)	2.8 ± 1.6	2.7 ± 1.5	0.852	0.394	−0.106	0.269
Perceived learned during lockdown (1–5)	3.0 ± 1.2	3.1 ± 1.17	−0.892	0.373	−0.210	0.079
Convenience to learn during lockdown (1–5)	3.4 ± 1.2	3.4 ± 1.1	0.008	0.994	−0.142	0.144
Motivation to learn during lockdown (1–5)	2.8 ± 1.2	2.8 ± 1.2	−0.088	0.930	−0.152	0.139
Difficulty to learn during lockdown (1–5)	3.0 ± 1.1	3.0 ± 1.1	−0.351	0.725	−0.163	0.113
Demanding activities during lockdown (1–5)	3.2 ± 1.1	3.1 ± 1.1	1.694	0.091	−0.018	0.249
Preferred learning method during lockdown (1–5)	2.6 ± 1.3	2.7 ± 1.4	−0.131	0.896	−0.175	0.153
Grades during lockdown (1–5)	3.5 ± 1.1	3.4 ± 1.1	0.308	0.758	−0.113	0.155
Perceived learned during hybrid phase (1–5)	3.1 ± 1.4	3.2 ± 1.2	−0.88	0.379	−0.234	0.089
Convenience to learn during hybrid phase (1–5)	3.1 ± 1.4	3.1 ± 1.2	−0.608	0.543	−0.213	0.112
Motivation to learn during hybrid phase (1–5)	2.9 ± 1.4	3.0 ± 1.3	−1.277	0.202	−0.275	0.058
Difficulty to learn during hybrid phase (1–5)	2.6 ± 1.3	2.8 ± 1.2	−2.773	0.006	−0.381	−0.065
Demanding activities during hybrid phase (1–5)	2.9 ± 1.4	3.2 ± 1.2	−3.185	0.001	−0.419	−0.099
Preferred learning method during hybrid phase (1–5)	2.8 ± 1.5	2.8 ± 1.4	−0.880	0.379	−0.264	0.100
Grades during hybrid phase (1–5)	3.1 ± 1.4	3.2 ± 1.3	−1.464	0.143	−0.293	0.043
Perceived learned during face-to-face phase (1–5)	3.3 ± 1.6	3.3 ± 1.5	−0.255	0.799	−0.219	0.169
Convenience to learn during face-to-face phase (1–5)	3.0 ± 1.6	3.1 ± 1.5	−0.585	0.558	−0.250	0.135
Motivation to learn during face-to-face phase (1–5)	3.0 ± 1.6	3.1 ± 1.6	−0.775	0.439	−0.274	0.119
Difficulty to learn during face-to-face phase (1–5)	2.5 ± 1.5	2.6 ± 1.4	−2.092	0.037	−0.363	−0.012
Demanding activities during face-to-face phase (1–5)	2.9 ± 1.5	3.1 ± 1.5	−1.700	0.089	−0.346	0.025
Preferred learning method during face-to-face phase (1–5)	3.0 ± 1.7	3.2 ± 1.6	−1.244	0.214	−0.337	0.075
Grades during face-to-face phase (1–5)	3.0 ± 1.6	3.1 ± 1.5	−0.471	0.638	−0.236	0.145
Preferred learning method General (1–5)	1.8 ± 1.1	1.7 ± 1.0	0.913	0.362	−0.072	0.196

**Table 2 ijerph-19-14729-t002:** Differences between gender of the psychological profile of students.

Variable	Female	Male	t	*p*	95% Confidence Interval	
					Lower	Upper
STAI (1–4)	15.0 ± 4.4	13.6 ± 4.3	5.164	0.000	0.862	1.919
UCLA (1–3)	5.6 ± 2.0	5.2 ± 2.0	2.623	0.009	0.082	0.567
PSS-4 (0–4)	8.1 ± 3.1	7.0 ± 3.2	5.550	0.000	0.701	1.468

(STAI) State–Trait Anxiety Inventory scale; (PSS-4) perceived stress scale; (UCLA) loneliness scale. Differences between genders (*p* < 0.05).

## Data Availability

All the data are presented in the study.

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
