# Peer review of "Gender Differences of University Students in the Online Teaching Quality and Psychological Profile during the COVID-19 Pandemic"

_ijerph, 2022, doi:10.3390/ijerph192214729_

Round 1

Reviewer 1 Report

Hi,

The manuscript needs enhancement. So please find the attached review report for more information to improve the paper.

Regards

Author Response

Dear Reviewer,

First of all, thank you for your work and effort in this review.We have followed your indications and have made the appropriate changes.We hope that these changes will improve the quality of the document.

Best regards,

----------------------------------------------------------------------------

Gender Differences of University Students in the Online 2 Teaching Quality and Psychological Profile during the COVID- 19 Pandemic After reviewing the article titled " Gender Differences of University Students in the Online 2 Teaching Quality and Psychological Profile during the COVID-19 Pandemic", I conclude that the results are worthy of being published. However, the manuscript has to be improved with major revisions.

Thank you very much for the positive appreciation regarding the document, we hope that the major revisions that we have carried out and indicated in yellow in the text, answered in this document, have improved the final document and therefore its quality.

Thank you,

S.Sato

General The article would benefit from language editing as there are minor spelling errors and inconsistencies in the use of words/phrases. Few examples: In the abstract, the sentence, “Data suggest that during the lockdown phase and online teaching ….. (line 23) is not clear so please rephrase it. Also Preferred learning method independent……….. ? line 30-31.

The document has been completely revised, the various grammatical errors have been corrected with tools such as Grammarly, finally one of our bilingual colleagues in English, revised the document.

Three phases identified in this study are Virtual, Hybrid, and Face-to-face. Kindly ensure to use these phases consistently in the paper rather than rewording it.

The term virtual was used instead of online, especially in the introduction, it has been modified throughout the text.

Under section 2.3 these phases are explained with different terminologies ( lines 126 -129). Spelling error - undergraduate o graduate program (line no 107)

Thanks for the appreciation, marked in yellow in the text and modified.

Different font sizes (lines 151-166) Different font sizes (lines 274-275)

We believe that this may be due to the MPDI system, since this error does not appear in the original document.

Introduction The research problem and the literature gap are not explicitly stated. Hence, the article might benefit from including the research problem and the gap identified. A literature review to support the research gap and the aim of the study should be considered.

From lines 42 to 56, we indicate the different educational steps that were followed in the educational sector during the pandemic. A posteriori in lines 70 to 79 we indicate in reference to previous scientific literature the complications found at the level of quality of education and mental health.

We believe that the background is sufficiently justified, even so to close the introduction we have included a new paragraph that we believe justifies the hypothesis and development of this research.

Materials and methods This section needs a major restructuring as the content in this section has no logical flow and is not strong enough to support the study's aim. Kindly address the following issues: The article is lacking an explanation of the research design.

This part has been rewritten, marked in yellow in the text. Thanks for the contribution.

Clearly state the research design and the justification for using the stated research design. Lack of explanation on sampling method, data collection method, and response rate, descriptive statistics of respondents. Sections 2.1 – 2.5 are not logically connected. So please choose an appropriate title for each section and rearrange the contents accordingly.

We have followed the structure of articles already published on the same topic and journal, based on questionnaires. We attach the references in case you want to check the methodological structure.

Rodriguez-Besteiro, S., Tornero-Aguilera, J. F., Fernández-Lucas, J., & Clemente-Suárez, V. J. (2021). Gender differences in the COVID-19 pandemic risk perception, psychology, and behaviors of Spanish university students. International Journal of Environmental Research and Public Health, 18(8), 3908.

Rodriguez-Besteiro, S., Valencia-Zapata, G., Beltrán de la Rosa, E., & Clemente-Suárez, V. J. (2022). Food consumption and COVID-19 risk perception of university students. Sustainability, 14(3), 1625.

What theoretical model or framework is the foundation for the variables identified for measuring the perception of academic quality? Please clearly state and justify. It would be better to measure the reliability and validity of the instrument used for measuring the perception of academic quality.

We have referenced in the text the methodology used and followed as in previous works.

Please state the motivation for using the STAI – State-Trait Anxiety Inventory scale, 160 UCLA - loneliness scale and PSS - perceived stress scale with proper literature support.

References 24, 25 and 26, all included in the text.

 Include the justification for using the Kolmogorov-Smirnov test with proper reference.

We have referenced in the text the methodology used and followed as in previous works.

Page 2 of 2 Results This section needs to be more organized. The stats provided for the Health sciences (45%), Social sciences (39%), and the remaining 16% belong to which field? Kindly organize the Table 1 contents based on the three identified phases ( virtual, hybrid, and faceto-face). Provide the details of the findings more clearly.

             Has been modified

Explicitly indicate the significant p-values and provide a proper footnote. The obtained results need to be explained more. The footnote provided for Table 2 is not clear. Please follow the proper footnote style

             This is included either in the footnote and in the main table. Marked in yellow

Discussion Kindly update the discussion, according to the changes in “methods and materials” and results.

             Done

Please check the sentence in lines (176-177). Is it “perception of quality” or “perception,quality”? please be consistent with the use of perception of academic quality.

             Many thanks! Has been addressed and marked in yellow

The important findings would be more clearly focused if the analysis and presentation of the data were anchored to the chosen study design in addition to being reader-friendly.

After the discussion, a new and easier to read version is offered in the Practical Applications and Conclusions.

Therefore it would be preferable to align the methods and materials, results, discussion and conclusion. References There is no consistency in the references so strongly recommend being consistent with the reference style and also translating non-English references into English.

We have used a bibliographic manager for it, and we have rechecked it, it should be in order now

Reviewer 2 Report

pp. 1, line 23-26 - This sentence is not entirely clear: „Data suggest that during the lockdown phase and online teaching, students presented higher levels of stress, class attendance, convenience to learn, difficulty in learning and lower grading score, motivation, perceived learned, and preferred learning method, compared to other phases of teaching.

Students presented higher levels of „class attendance“? „convenience to learn“? It feels like some crucial information is missing. Also „perceived learned“? Learned what? Please make it clearer.

pp. 2, „In the case of university institutions, during the academic years 20/21 and 21/22, universities have proposed alternatives to the face-to-face model to an emerging virtual modality and have forced an emergency readaptation in all educational stages. Universities started in a synchronous format, that is, with professors and students meeting at the same time in classes through a virtual platform, the practical activities were postponed or substituted when they were possible. In the first six months, almost all the classes switched to this 100% online modality (8).“ – This part of the text is written as if the same procedure existed in all the universities of the world. I wonder if you have this information for all universities? Maybe it would be better to distance yourself from it and indicate what you are talking about. Is it about your national context?

Please think about the sentence structure. For example „impact of mental health on students and professors“ (pp. 2, line 83-84), did you mean the impact of the pandemic on the mental health of students and professors?

pp. 2-3, line 95-98 - „Research in this field has been able to confirm that significant correlations can be established between gender differences and social determinants in relation to the appearance of mental illnesses. Gender differences and their consequences on mental well-being can be observed in all countries of the world.“ - Please provide the references of the works in which this information appears. This is necessary if you make such strong statements.

pp. 3, line 100-102 „the gender differences of university students in online teaching on the perception of quality and psychological profile during the COVID-19 pandemic“ – This is perhaps the most important sentence of the whole manuscript, because it tells us concretely something about the aim of your research. Unfortunately, it is not clear enough. It is possible that it is a confusion caused by the linguistic structure of the sentence. I wonder if the aim of this manuscript is to examine gender differences in the perception of quality and psychological profile of university students during the pandemic COVID -19?

pp. 3, line 132 - „...Likert scale, where 1 means the lowest and 5 the highest.“ The lowest what, and the highest what? Incomplete information.

pp. 4, line 162-164 – „Furthermore, the male students presented higher values in the results of difficulty learning and demanding activities during the hybrid phase and difficulty learning during the face-to-face phase.“ – Can you please explain what are demanding activities and difficulty learning? It has not been explained anywhere in the text so far, and if the difference between the genders in these variables is already known, it would be useful to explain what it refers to. What is more difficult for men to bear than for women?

pp. 6, line 176 – You mention the hypothesis, but you did not present it earlier in the paper. If there is a hypothesis, then it must be clearly stated according to the goal and tasks of the research.

Necessary linguistic refinement of the work. Important information is lost due to poor sentence formation.

Author Response

Dear Reviewer,

First of all, thank you for your work and effort in this review.
We have followed your indications and have made the appropriate changes.
We hope that these changes will improve the quality of the document.

Best regards,

----------------------------------------------------------------------------------

  1. 1, line 23-26 - This sentence is not entirely clear: „Data suggest that during the lockdown phase and online teaching, students presented higher levels of stress, class attendance, convenience to learn, difficulty in learning and lower grading score, motivation, perceived learned, and preferred learning method, compared to other phases of teaching.“

             It has been modified and marked in yellow, many thanks.

Students presented higher levels of „class attendance“? „convenience to learn“? It feels like some crucial information is missing. Also „perceived learned“? Learned what? Please make it clearer.

             It has been modified and marked in yellow, many thanks.

  1. 2, „In the case of university institutions, during the academic years 20/21 and 21/22, universities have proposed alternatives to the face-to-face model to an emerging virtual modality and have forced an emergency readaptation in all educational stages. Universities started in a synchronous format, that is, with professors and students meeting at the same time in classes through a virtual platform, the practical activities were postponed or substituted when they were possible. In the first six months, almost all the classes switched to this 100% online modality (8).“ – This part of the text is written as if the same procedure existed in all the universities of the world. I wonder if you have this information for all universities? Maybe it would be better to distance yourself from it and indicate what you are talking about. Is it about your national context?

Thanks for the appreciation, we missed including the term Ibero-American countries, since in this framework and context, which is the one we are studying, the procedure that the universities adopted was the same. At different time stages, but the procedure was similar. It is also referenced.

Please think about the sentence structure. For example „impact of mental health on students and professors“ (pp. 2, line 83-84), did you mean the impact of the pandemic on the mental health of students and professors?

             Indeed, many thanks it has been modifyed and marked in yellow in the text

  1. 2-3, line 95-98 - „Research in this field has been able to confirm that significant correlations can be established between gender differences and social determinants in relation to the appearance of mental illnesses. Gender differences and their consequences on mental well-being can be observed in all countries of the world.“ - Please provide the references of the works in which this information appears. This is necessary if you make such strong statements.

             Indeed, has been modified.

  1. 3, line 100-102 „the gender differences of university students in online teaching on the perception of quality and psychological profile during the COVID-19 pandemic“ – This is perhaps the most important sentence of the whole manuscript, because it tells us concretely something about the aim of your research. Unfortunately, it is not clear enough. It is possible that it is a confusion caused by the linguistic structure of the sentence. I wonder if the aim of this manuscript is to examine gender differences in the perception of quality and psychological profile of university students during the pandemic COVID -19?

We have modifed it and a new paragraph has neen included. Reviwer 1 also made the same statement. Many thanks

  1. 3, line 132 - „...Likert scale, where 1 means the lowest and 5 the highest.“ The lowest what, and the highest what? Incomplete information.

             Adressed.

  1. 4, line 162-164 – „Furthermore, the male students presented higher values in the results of difficulty learning and demanding activities during the hybrid phase and difficulty learning during the face-to-face phase.“ – Can you please explain what are demanding activities and difficulty learning? It has not been explained anywhere in the text so far,

             Marked in yellow! Thanks we have adressed it.

  1. 6, line 176 – You mention the hypothesis, but you did not present it earlier in the paper. If there is a hypothesis, then it must be clearly stated according to the goal and tasks of the research.

Lines 191-195 in the discussion, in where we confirm. Previosuly, it was stated in a new paragraph in lines 100 and 108.

Round 2

Reviewer 2 Report

Thank you for including my suggestions. Best regards